# Improved YOLOv7 Network Model for Gangue Selection Robot for Gangue and Foreign Matter Detection in Coal

**DOI:** 10.3390/s23115140

**Published:** 2023-05-28

**Authors:** Dengjie Yang, Changyun Miao, Xianguo Li, Yi Liu, Yimin Wang, Yao Zheng

**Affiliations:** 1School of Mechanical Engineering, Tiangong University, Tianjin 300387, China; 2Tianjin Photoelectric Detection Technology and System Key Laboratory, Tiangong University, Tianjin 300387, China; 3School of Electronics and Information Engineering, Tiangong University, Tianjin 300387, China; 4Center for Engineering Internship and Training, Tiangong University, Tianjin 300387, China

**Keywords:** gangue, foreign matter, identification and detection, YOLOv7 network model, YOLOv71 + COTN network model, gangue selection robot

## Abstract

Coal production often involves a substantial presence of gangue and foreign matter, which not only impacts the thermal properties of coal and but also leads to damage to transportation equipment. Selection robots for gangue removal have garnered attention in research. However, existing methods suffer from limitations, including slow selection speed and low recognition accuracy. To address these issues, this study proposes an improved method for detecting gangue and foreign matter in coal, utilizing a gangue selection robot with an enhanced YOLOv7 network model. The proposed approach entails the collection of coal, gangue, and foreign matter images using an industrial camera, which are then utilized to create an image dataset. The method involves reducing the number of convolution layers of the backbone, adding a small size detection layer to the head to enhance the small target detection, introducing a contextual transformer networks (COTN) module, employing a distance intersection over union (DIoU) loss border regression loss function to calculate the overlap between predicted and real frames, and incorporating a dual path attention mechanism. These enhancements culminate in the development of a novel YOLOv71 + COTN network model. Subsequently, the YOLOv71 + COTN network model was trained and evaluated using the prepared dataset. Experimental results demonstrated the superior performance of the proposed method compared to the original YOLOv7 network model. Specifically, the method exhibits a 3.97% increase in precision, a 4.4% increase in recall, and a 4.5% increase in mAP0.5. Additionally, the method reduced GPU memory consumption during runtime, enabling fast and accurate detection of gangue and foreign matter.

## 1. Introduction

In recent years, the integration of robots in gangue and foreign matter screening has gained significant momentum as a notable development trend. Researchers have shown considerable interest in the visual information-based gangue robot selection method [1,2,3]. The efficacy of a gangue selection robot, which relies on visual information, is contingent upon precise gangue and foreign object recognition, as well as the speed of the manipulator. Currently, gangue selection robots face challenges, including slow selection speed and limited recognition accuracy. Attaining high-speed and accurate target recognition is a pivotal factor in enabling the automatic screening of a visual information-based gangue selection robot [4].

In the field of target recognition, researchers are progressively transitioning from relying on physical differences to exploiting faster and more intuitive visual distinctions. The integration of machine learning techniques, especially deep learning, has brought about a revolutionary change in traditional image classification recognition [5]. These methods demonstrate high accuracy in controlled laboratory settings. However, they suffer from limitations due to the manual extraction of image features inherent in traditional machine learning approaches, resulting in models with limited generalization capabilities. Additionally, the recognition speed of conventional machine learning methods is insufficient to meet the practical production requirements of coal mines.

To overcome the limitations of slow speed, low efficiency, fluctuating accuracy, and limited applicability associated with traditional image processing-based classification and recognition methods, recent research has delved into advanced techniques. Gao et al. [6] tackled these challenges by training and evaluating collected images using the U-net network, resulting in successful localization and classification of coal gangue. Li et al. [7] introduced a hierarchical coal gangue detection framework based on deep learning models. enabling the identification of multiple targets within a single image. They progressively refined their modules, transitioning from a simple LeNet structure to an Inception structure [8,9,10], and eventually to a ResNet structure [11,12]. The depth of the network modules and the number of corresponding layers increased throughout the research, starting from the 8 convolutional layers of the LeNet structure and culminating in the deep network structure of ResNetSt-269, which comprised 269 layers [13].

The continuous optimization of network model designs has resulted in significant improvements in recognition accuracy and speed. These advancements have not only enhanced the models’ recognition accuracy and generalization ability [14], but they have also addressed various challenges in image classification, image segmentation [15], and super-resolution [16]. Notable contributions include the development of target detection algorithms, such as R-CNN [17], fast R-CNN [18], R-CNN [19], as well as the SSD series [20,21] and YOLO series [22,23,24,25]. Attention mechanisms, such as SENet [26], CBAM [27], and ECA [28,29], have also played a significant role in further enhancing performance.

As network models continue to improve in speed and accuracy, the size of network parameters is also increasing. This presents challenges in practical applications, particularly in the context of the coal gangue recognition classification network based on Inception module migration learning, where the large memory requirements, reaching gigabytes, impede the embedding process. To tackle this issue, Google introduced the MobileNet network model [30,31]. By utilizing depth-separable convolution in place conventional convolution operations, the overall network parameters are reduced. However, concerns still exist regarding lower recognition accuracy.

This study addresses the issues of slow gangue selection speed and low recognition accuracy in gangue selection machine methods. To overcome these challenges, an improved gangue and foreign matter recognition network model is proposed. Anchor rods and I-beams were selected as the representative foreign matter. The YOLOv7 model structure served as the foundation for this approach. An industrial camera was utilized to capture raw images of coal blocks, gangue blocks, anchor rods, and I-beams. Data augmentation techniques were employed to enhance the dataset’s representativeness and alignment with coal mine environments. The improved model incorporated a new contextual transformer networks (COTN) module, which replaced the convolution in ResNet, resulting in the new YOLOv71 + COTN model. Additionally, heavy parameter processing was applied to the output section of the new model. These enhancements significantly improved network model performance, including enhanced output feature expressiveness, recognition speed, and accuracy, while minimizing the increase in model parameter volume and complexity.

In this study, the distance intersection over union (DIoU) loss function was chosen for border regression to effectively handle the overlap between the predicted frame and the ground truth frame. The utilization of the DIoU loss function offered several advantages, including improved robustness, faster convergence speed, and enhanced detection performance. The prediction frame was represented as (*x*_1_, *x*_2_, *x*_3_, *x*_4_), while the ground truth frame was represented as (*x*_1_’, *x*_2_’, *x*_3_’, *x*_4_’). By employing the DIoU loss, the predicted frame was less affected by the size of the ground truth bounding box, resulting in improved robustness of the model. Unlike *l*_2_ loss, DIoU loss directly minimized the distance between the two bounding boxes, leading to faster convergence and better detection performance. Furthermore, DIoU loss function was scale-invariant, further contributing to its effectiveness.

Prior to training the model on the dataset, the input sample images underwent a normalization process. This step played a vital role in enhancing the accuracy and efficiency of gangue and foreign matter image detection, thereby endowing the new model with robust multi-scale image processing capabilities. As a result, the new model demonstrated rapid detection and identification of gangue and foreign matter within coal samples. Furthermore, even in scenarios where these objects are covered with coal dust, they can still be reliably identified based on their distinctive physical characteristics. The new and improved model achieved these capabilities while reducing the number of parameters and the complexity of the network structure, leading to enhanced speed and accuracy in object identification. These advancements offer the benefits of low detection costs, high accuracy, and reliability.

## 2. Related Works

The presence of gangue and foreign matter in coal production has a substantial impact on the heating value of coal and can cause damage to transportation equipment, such as belt conveyors. Thus, it is essential to detect and effectively remove gangue and foreign matter to ensure reliable, efficient, and environmentally friendly sorting processes [32]. The development of automatic systems for accurate and efficient detection and removal of gangue and foreign matter holds paramount practical significance in the coal industry.

Traditional methods for gangue selection in coal processing include manual gangue selection, jigging gangue selection, heavy media gangue selection, and ray gangue selection [33]. Manual gangue separation is characterized by high labor intensity and low efficiency. Jigging gangue selection offers a simple process, ease of operation, and good processing capacity. However, it struggles with smaller gangue blocks. Heavy medium gangue exhibits high efficiency and wide applicability to different gangue types, but it suffers from a complex process system, significant equipment wear, and high cost. Ray gangue selection involves X-ray and γ-ray techniques, providing high efficiency and reliability. Nevertheless, environmental concerns arise with this method, which makes it less aligned with the principles of green development.

The YOLOv7 network is composed of three main parts [34]: the input, backbone, and head, as shown in Figure 1. In contrast to YOLOv5, YOLOv7 combines the neck and head layers into a head layer, while retaining the functionality of each part similar YOLOv5. The backbone is responsible for extracting features, and the head is utilized for making predictions. The network process in YOLOv7 begins with the preprocessing of the input image into a standardized RGB format with dimensions of 640 × 640. The processed image is then fed into the backbone network. According to the outputs of the three layers within the backbone network, feature maps of different sizes are generated and subsequently passed through the head layer. Following the RepVGG block and Conv, target detection takes place, ultimately yielding the final output.

The backbone part of the YOLOv7 network model comprises four CBSs (i.e., convolution, batch normalization, and Sigmoid weighted liner unit (SiLU) activation) followed by an enhanced light-weight aggregation network (ELAN). By combining the outputs of three MaxPooling (MP) layers with ELAN, feature maps of sizes 80 × 80 × 512, 40 × 40 × 1024, and 20 × 20 × 1024 are obtained. Each MP layer consists of 5 layers, while ELAN is composed of 8 layers, resulting in a total of 51 layers in the backbone section. The head section of YOLOv7 adopts a structure similar to pafpn observed in YOLOv4 and YOLOv5. Initially, the final output of the 32-fold downsampled feature map C5 from the backbone is obtained. Subsequently, spatial pyramid pooling with convolutional-spatial path (SPPCSP) is applied to reduce the number of channels from 1024 to 512.

## 3. Methods

### 3.1. YOLOv7 Network Model Improvement

The YOLOv7 backbone feature extraction network is a CNN-based network known for its translation invariance and localization capabilities. However, it lacks the ability to model globally and over long distances. To overcome this limitation, the transformer framework, widely used in natural language processing, is introduced to construct the CNN + transformer architecture, forming the COTN module. By integrating the transformer framework, the target detection capabilities are significantly enhanced, especially for detecting small gangue blocks, coal blocks, and densely packed objects on the conveyor. This adaptation is vital to effectively handle the presence of a large number of coal blocks on the conveyor belt. Furthermore, insights from other advancements in the YOLO model are incorporated to further improve the overall performance [35,36,37].

The introduced transformer framework, implemented in this study, incorporates a novel spatial modeling mechanism based on dot product self-attention. This framework leverages recursive gated convolution and enables higher-order spatial interactions through gated convolution and recursive design. Consequently, it offers a high degree of flexibility and customizability. YOLOv7 model’s substantial number of parameters and computational resources enable the compatibility of the CNN + transformer architecture with various convolutional variants. Furthermore, it allows for the extension of second-order interactions in self-attention to arbitrary orders without incurring additional computational overhead.

The improvements to the YOLOv7 model are depicted in Figure 2, visually illustrating the enhancements. Specifically, the initial segment of the backbone enhanced performance achieved by reducing the number of convolutional layers by half. This reduction leads to a decrease in the overall number of network layers and the spatial size of the model, resulting in a shorter model running time. Consequently, these modifications give rise to the YOLOv71 network structure. Table 1 presents a comprehensive comparison of the number of layers and running time between the two model structures.

To enhance the learning ability of small target detection, a small-sized detection layer is introduced in the head section. The inclusion of the COTN module further enhances the capabilities of the model. The DIoU loss border regression loss function is employed to calculate the overlap between the predicted frame and the ground truth frame, enabling the identification of gangue and foreign objects based on this overlap. Additionally, a dual-path attention mechanism is incorporated to improve recognition accuracy.

In this study, it was observed that approximately 70% of the gangue particles had a size greater than 50 mm, indicating a prevalence of large gangue. The remaining 30% of the gangue particles had a size less than 50 mm, representing the category of small gangue. The coal mine environment poses challenges in detecting smaller objects during target detection. To account for the variations in particle sizes encountered in real-world scenarios, the gangue used in the experiment encompassed both large and small gangue.

This study presents the process of gangue and foreign matter detection in coal using an improved YOLOv7 network model for a gangue selection robot, as shown in Figure 3. An industrial camera was used to capture images of coal, gangue, and foreign matter. These images underwent classification, annotation, and augmentation to generate a dataset. The dataset was then utilized to train and test the improved YOLOv7 network model, enabling the determination of optimal model weights. Subsequently, the obtained model weights were applied to the improved YOLOv7 network model to facilitate the detection and identification of gangue and foreign objects in coal on a belt conveyor.

### 3.2. Re-Parameterization

Two convolutional layers were incorporated into the batch normalization (BN) layer in the head section. This reparameterization of the three components is illustrated in Figure 4.

The convergence equation for Conv and BN can be expressed as follows:(1)$$New\;Conv=\frac{r\left(\left(wx+b\right)-m\right)}{v}+\beta =\frac{rw}{v}x+\frac{r(b-m)}{v}+\beta$$
where *w* denotes the convolutional bias, *b* denotes the convolutional bias, *r* and *β* denote the parameters that can be learned in BN, *m* denotes the input mean in BN, and *v* denotes the input standard deviation in BN.
(2)$$Ŵ=\frac{rw}{v}$$
(3)$$b=\frac{r(b-m)}{v}+\beta$$

Equations (2) and (3) are combined and incorporated into Equation (1), resulting in the derivation of the new fusion Equation (4).
(4)New Conv=ŵx+b

The YOLOv7 network model structure consists of 415 layers, contributing to a significant number of network layers. However, this extensive layer count led to prolonged training and recognition time, creating challenges in achieving fast recognition of foreign matter in coal on a rapidly moving belt conveyor. To tackle this issue, a reparameterized convolution process was introduced before image output. This process enhanced the model’s running speed, resulting in improved recognition speed while maintaining consistent model performance.

### 3.3. CONT Module

COTN utilized the transformer framework to replace the convolution in ResNet, serving as the backbone of the network. This replacement involves the utilization of a 1 × 1 convolution, enabling the seamless integration of contextual information mining and self-attentive learning within a unified architecture. Through the enhancement of self-attention, COTN enables efficient learning of contextual information, leading to improved expressiveness of the output features.

While the transformer framework demonstrates strong global modeling capability for long-distance interactions, it primarily calculates the attention matrix based on the interaction between query and key, neglecting the connection between adjacent keys. To overcome this limitation, a 3 × 3 convolution was applied to the key to model static contextual information. This convolution operation, illustrated in Figure 5, captures localized information. The key was combined with the modeled query and context information using the COTN module. Following this, two consecutive 1 × 1 convolutions were employed to generate dynamic contexts through self-attention. Finally, the static and dynamic context information was fused together to produce the output.

### 3.4. Margin Regression Loss Function

In the target detection process, the target bounding box was commonly represented by four variables (i.e., *x*, *y*, *w*, *h*). In this study, the predicted frame was denoted as (*x*_1_, *x*_2_, *x*_3_, *x*_4_) and the ground true frame was denoted as (*x*_1_’, *x*_2_’, *x*_3_’, *x*_4_’). Notably, the intersection over union (IoU) loss treats both large and small bounding boxes equally. When dealing with images of varying resolutions (i.e., different bounding box sizes), the prediction box obtained by *l*_2_ loss is more influenced by the size of the ground true bounding box. Conversely, the prediction box obtained with IoU loss is less affected by the ground true bounding box’s size, leading to improved robustness. Compared to *l*_2_ loss, IoU loss directly measures the distance between two bounding boxes, resulting in faster convergence speed and enhanced detection performance. Additionally, IoU loss demonstrates scale invariance, meaning it performs consistently well for both large and small bounding boxes.

This natural normalized loss function enhances the model’s ability to handle multiscale images. IoU value is 1 when the prediction perfectly matches the ground truth, and 0 when there is no overlap between the prediction and the ground truth. As IoU loss tends to approach positive infinity when IoU is 0, it decreases monotonically from positive infinity to 0 as IoU increases within the interval [0, 1]. IoU loss can be viewed as a form of cross-entropy loss of that quantifies the dissimilarity between the predicted and ground truth bounding boxes. Compared with *l*_2_ loss, which measures the Euclidean distance between bounding boxes, IoU loss directly captures the spatial overlap between the two bounding boxes, leading to faster convergence and improved detection performance. Additionally, IoU loss demonstrates scale invariance and remains effective for both large and small bounding boxes.

*l*_2_ loss border regression loss function can be calculated as follows:(5)$$l_{2}\;\mathrm{loss}={\left\|Prediction-Truth\right\|}_{2}^{2}$$

IoU loss border regression loss function can be calculated as follows:(6)$$IoU\;loss=-In\frac{Intersection(Prediction,Truth)}{Union(Prediction,Truth)}$$
where *Prediction* denotes the prediction box for the detection target, and *Truth* denotes the detection target truth box.

In situations where the prediction frame and the target frame do not overlap, the IoU loss function IoU(A,B) = 0, which does not capture the spatial distance between the two frames A and B. Consequently, the IoU loss function lacks differentiability and fails to effectively optimize cases where the frames do not intersect. Additionally, assuming fixed sizes for the prediction box A and target box B, the IoU value remains unchanged regardless of the specific intersection pattern between the boxes. Thus, IoU values alone do not provide detailed insights into the nature of the intersection. To overcome these limitations, the DIoU loss function was employed as a border regression loss. DIoU loss considers the distance between the prediction and target boxes, offering a more comprehensive optimization measure.

Figure 6 illustrates the data calculations for the predicted and ground truth boxes in relation to each detection target. The red boxes represent the ground truth boxes, while the green boxes correspond to the predicted boxes generated by the model. In this study, anchor rods and I-beams are selected as representative foreign matter. When *Prediction* and *Truth* do not intersect, the IoU value was 0. However, this value does not capture the spatial distance between the two frames, rendering the loss function non-differentiable. As a result, IoU loss cannot effectively optimize scenarios where the boxes do not intersect. Additionally, when assuming fixed sizes for both *Prediction* and *Truth*, their IoU values remain the same regardless of the specific intersection pattern between the boxes. Consequently, IoU values fail to provide information about the actual characteristics of the intersection. To overcome these limitations, DIoU loss is introduced as a border regression loss function, as depicted in Figure 7.

DIoU loss can be calculated as follows:(7)$$L=1-IoU+R(B,B^{gt})$$
(8)$$R(B,B^{gt})=\frac{{\left\|b-b^{gt}\right\|}_{2}^{2}}{c^{2}}$$
where *B* denotes *Prediction* box, *B^gt^* denotes *Truth* box, *R(B*, *B^gt^)* denotes the added penalty term, and *c* denotes the square of the diagonal length of the minimum enclosing box *c*. The DIoU loss introduces the direct Euclidean distance between the two boxes as a penalty term, leading to a faster convergence rate compared to generalized intersection over union (GIoU) loss. Furthermore, DIoU loss takes into account the relative proportions of the rectangular boxes in the penalty term, which helps resolve the issue of box intersection between *Prediction* and *Truth*.

### 3.5. Attention Mechanism

The attention mechanism in artificial neural networks is inspired by the information acquisition behavior of the human brain. When humans gather information through their senses, the presence of numerous stimuli can create distractions, making it challenging to focus on the desired information. To solve this problem, the brain employs specialized processing units that facilitate effective processing and monitoring of relevant information. In the context of artificial neural networks, the attention mechanism aims to replicate this behavior by assigning specific weights to different targets during the feature extraction process. This allows the neural network to prioritize and emphasize important targets or regions of interest while suppressing or disregarding irrelevant or uninformative regions.

The attention mechanism, as shown in Figure 8, involves the output of a green square region denoted as D (t_x_, t_y_, t_1_). In this representation, the central coordinates are represented by t_x_ and t_y_, while half of the side lengths of the square region are represented by t_1_. The upper left coordinates are given by (t_x_ − t_1_) and (t_y_ − t_1_), and the lower right coordinates are given by (t_x_ + t_1_) and (t_y_ + t_1_). This square region serves as the core region that captures image category features under the attention mechanism. The attention extraction network is composed of two main parts. The first part, denoted as ‘a’, represents the features extracted by the attention mechanism. This component focuses on capturing important information within the region of interest. The second part, denoted as ‘c’, represents the features extracted through the convolution operations. This component is responsible for capturing features at different scales and resolutions within the image.

## 4. Experiments and Analysis of Results

### 4.1. Configuration of Experimental Environment

The models in this study were trained and tested using PyTorch open-source framework renowned for its flexibility and versatility. The experiments were conducted on a Windows server running Windows Server 2019 with Windows 10 as the 64-bit operating system. The central processing unit (CPU) employed was an Intel(R) Xeon(R) Platinum 8255C CPU @ 2.50 GHz 2.49 GHz. To expedite the training and testing processes, a graphics processing unit (GPU) was utilized. Specifically, NVIDIA Tesla T4 (8G) was chosen for its computational prowess. The deep learning environment was established using Conda, with the following specifications: Python version 3.9.0, torch version 1.10.0, torchvision version 0.11.0, Torchaudio version 0.10.0, and CUDA version 10.2. As shown in Table 2, these hardware and software configurations were thoughtfully selected to ensure optimal performance and compatibility throughout the training and testing phases of the models.

To comprehensively analyze the performance of YOLOv71 + COTN network model, two additional modules were introduced: simulated attention mechanism (simAM) and background object training (BOT). These modules played a crucial role in facilitating a longitudinal comparison during both the model training and testing phases. To provide a cross-sectional comparison, the dataset was utilized for training and testing Yolov5 model. The pre-training parameter settings configurations for each model are detailed in Table 3, which helps to establish a solid foundation for the subsequent experiments and evaluations.

### 4.2. Dataset Creation

In the testing phase, a variety of targets were selected, including coal blocks, gangue blocks, anchor rods, and I-beams. These targets were deliberately chosen to encompass a wide range of physical characteristics. Gangue blocks closely resemble coal blocks in their physical form, while anchor rods exhibit a slender shape and I-beams possess a flat surface. Furthermore, both anchor rods and I-beams are primarily composed of iron elements. To acquire the necessary images for testing, an industrial camera was employed. Each detection target was photographed, and the dataset was subsequently augmented using various techniques such as rotating, mirroring, panning, brightness adjusting, and Gaussian blurring. The efficacy of these augmentation techniques can be observed in Figure 9. The objective of this dataset augmentation process was to more accurately simulate the real-world conditions of each detection target within a coal mine environment and to enhance the dataset’s representativeness.

The targets in the images were manually annotated using the label open-source annotation tool. The dataset was organized into four categories: bolt, coal, gangue, and I-beam. Each target was assigned a numerical label: 0 for anchor, 1 for coal block, 2 for gangue block, and 3 for I-beam. The annotation process involved carefully marking the targets within the images to ensure accurate labeling based on human visual perception. The annotation results were saved in visual object classes (VOC) format, with the corresponding XML files stored in a predetermined folder. Great care was taken to label all visible targets while avoiding the inclusion of ambiguous ones. This approach was employed to prevent unannotated targets from being mistaken as negative samples, thus preserving the algorithm’s ability to effectively distinguish between positive and negative samples.

### 4.3. Analysis of Experimental Results

When the network structure of the YOLOv7 network model is fixed, the perceptual area of the network is predetermined. The resolution of the input image plays a critical role in determining the proportion of the perceptual area within the image. Higher-resolution images result in a lower percentage of the perceptual area, which reduces the effectiveness of capturing the local information for predicting objects at different scales. Consequently, this can lead to a decrease in detection accuracy. The variance in input image size has a significant impact on the model’s detection performance. The underlying network part often generates feature maps that are 10 times smaller than the original image. As a result, features related to small targets may not be adequately captured by the detection network, particularly for relatively small gangue blocks on the belt conveyor. To mitigate these challenges, the input image pairs in the experiments were resized to a standardized size of 640 × 640, as indicated in Table 4. It is important to note that the resized dataset images did not exceed the size of the original images in the dataset. This standardization of image size enhances the robustness of the detection model to variations in target size, to a certain extent.

The number of parameters in a network model refers to the total size, in bytes, of each network layer. It quantifies the amount of video memory occupied by the model. In this study, the spatial complexity of the model is evaluated based on the number of parameters, which provides insights into the model’s size and memory requirements. On the other hand, the time complexity of the model is evaluated using the amount of computation required. The computation volume measures the duration of the model’s detection process and is expressed as the number of floating-point operations per second. Additionally, the GPU runtime memory represents the amount of server space occupied by the model when running. It reflects the memory usage during the model’s execution on the GPU and is a crucial consideration for ensuring efficient utilization of server resources.

The YOLOv7 network model, derived from the YOLOv5 network model, exhibits a reduction in both network layers and parameters compared to the original YOLOv7 network model. Interestingly, despite having fewer layers and parameters, the YOLOv5 network model occupies an additional 3.93 G of space when compared to the YOLOv7 network model. Specifically, the YOLOv71 network model boasts 22 fewer layers and 1,701,083 fewer parameters. Furthermore, the memory usage by GPU is reduced by 0.15 G. To further analyze the operating parameters, additional modules such as simAM, BOT, and COTN were incorporated into the models. Among these modules, the COTN module demonstrated superior performance compared to the simAM and BOT modules.

Table 5 presents the detection results for each detection target in the new improved model, including accuracy, recall, and average accuracy mean. Upon analyzing the results in Table 4, it is evident that anchor rods achieve higher identification accuracy compared to coal and gangue targets. This difference in performance can be attributed to the similarity in shape and appearance between gangue and coal, posing challenges for accurate discrimination. Additionally, I-beam, characterized by a larger surface area is larger and a tendency to accumulate coal dust, visually resembles coal, thereby impeding precise recognition. Consequently, the recognition performance for gangue and I-beams is relatively poorer when compared to other targets.

The YOLOv71 + simAM network model demonstrates improved recognition specifically for coal when compared to the YOLOv71 network model. However, the overall detection results for each target in the YOLOv71 network model are lower. On the other hand, the YOLOv71 + BOT network model exhibits enhanced recognition capabilities specifically for anchor rods following the model’s improvements. Notably, the YOLOv71 + COTN network model demonstrates improvements in terms of accuracy, recall, and average accuracy for each detection target when compared to the YOLOv71 model.

Box metric represents the mean value of the loss function, where a smaller value indicates more accurate detection, Objectiveness metric represents the mean value of the target detection loss, with a smaller value indicating more accurate detection of targets. The classification metric represents the mean value of the classification loss, where a smaller value indicates a more accurate classification. The comparison results presented in Figure 10 illustrate that YOLOv71 + COTN network model achieves lower detection values on the training set compared to the other models. The data curve exhibits a relatively flat trend, with a general decrease in values. These findings indicate that the YOLOv71 + COTN model outperforms the other three models in terms of detection accuracy.

The precision metric represents the accuracy of positive predictions, while the recall metric represents the rate of correctly identified positive instances. The true metric represents the accuracy of the positive metric, representing the number of positive samples recalled, which describes the number of positive samples correctly recalled by the classifier. It provides insights into how well the classifier identifies positive instances. The comparison results depicted in Figure 11a,b show that YOLOv71 + COTN network model exhibits higher accuracy and recall values compared to the other models during training. These results suggest that YOLOv71 + COTN model achieves better recognition performance, with improved accuracy in identifying positive instances and a higher recall rate for true positive examples.

mAP0.5 metric represents the mean accuracy values at a threshold of 0.5. mAP0.5:0.75 metric represents the mean accuracy values for thresholds ranging from 0.5 to 0.75, with an interval of 0.05. Upon comparing the information presented in Figure 11c,d, it is evident that the YOLOv71 + COTN network model achieves the highest mean accuracy for the detection of gangue and foreign matters at the specified threshold.

The simAM, BOT, and COTN modules were integrated into the YOLOv71 network model to conduct ablation experiments and evaluate their impact on performance. The test results for each network model are shown in Table 6. It is evident that the network model with the additional module exhibits notable improvements in accuracy and recall compared to the YOLOv7 network model. Among the three modules, the COTN module is the most effective, achieving an impressive identification accuracy of 91.3%.

Figure 12 provides a visual representation of the detection results obtained by each network model for the same target. The analysis of the model output images reveals a clear pattern: As the network model undergoes continuous improvement, there is a noticeable enhancement in the accuracy and reliability of target detection. Of particular note is the performance of the new YOLOv71 + COTN model, which achieves the highest confirmation rate for the target, reaching an impressive value of 90% when the target is clearly visible.

Figure 13 represents the model’s performance in identifying targets during the testing phase. Each target is assigned a numerical label: 0 for anchor, 1 for coal, 2 for gangue, and 3 for I-beam. Figure 13 reveals that both YOLOv7 and YOLOv71 network model exhibit recognition errors in the randomly outputted detection images. Specifically, the YOLOv7 network model has two instances of recognition errors, while the YOLOv71 network model has one instance of recognition error. However, with the incorporation of the new module, the network model demonstrates a notable absence of recognition errors.

## 5. Conclusions

This study presents a novel approach for detecting gangue and foreign matter in coal using an improved YOLOv7 network model. The improved YOLOv7 network model is specifically tailored to achieve accurate and reliable detection of gangue and foreign matter in coal samples. To enhance the performance of YOLOv7 network model, several key improvements were introduced. Firstly, the number of convolutional layers in the backbone was halved, resulting in a more streamlined and efficient detection process. This reduction in convolutional layers not only accelerated the detection speed but also enhanced the overall detection efficiency for gangue and foreign matter. Furthermore, small-size detection layers were incorporated into the head. This addition aimed to enhance the model’s ability to detect and classify small targets more effectively. By specifically focusing on small-sized objects, the model could improve its accuracy in identifying and localizing these challenging targets. Additionally, the COTN module further enhanced the model’s detection accuracy. This module leverages innovative techniques to refine the model’s feature representation, ultimately improving its ability to discriminate between different classes of gangue and foreign matter. To calculate the overlap between the predicted and real frames, DIoU loss border regression loss function was employed. This loss function provided a comprehensive measure of the intersection between the predicted and ground truth bounding boxes. By considering both the spatial distance and proportional characteristics of the boxes, the model could effectively identify gangue and foreign matter based on this calculated overlap. To improve recognition accuracy, a dual-path attention mechanism was integrated into the model. This mechanism allowed the model to selectively focus on relevant features while suppressing irrelevant or distracting information. By effectively attending to important regions of the input, the model achieved enhanced recognition accuracy and robustness.

This research proposes an improved the YOLOv7 network model specifically designed for the identification of gangue and foreign matter in coal, with a focus on its application in gangue sorting robots. The proposed model, YOLOv71 + COTN, was trained and evaluated using a dedicated dataset to ensure accurate identification of gangue and foreign matter. The experimental results revealed significant improvements compared to YOLOv71 + COTN network model approach. This proposed network model demonstrated notable enhancements across various performance metrics. Precision was improved by 3.97%, recall was increased by 4.4%, and mAP0.5 was improved by 4.5%, compared to the YOLOv7 network model approach. By reducing the number of parameters, the model’s efficiency was optimized, resulting in reduced memory requirements on GPU during operation. This reduction in memory consumption contributes to improved gangue selection speed and recognition accuracy, further enhancing the overall performance of the model. As a result, the YOLOv71 + COTN network model is well-suited for integration in belt conveyor gangue sorting robots.

## Figures and Tables

**Figure 1 sensors-23-05140-f001:**
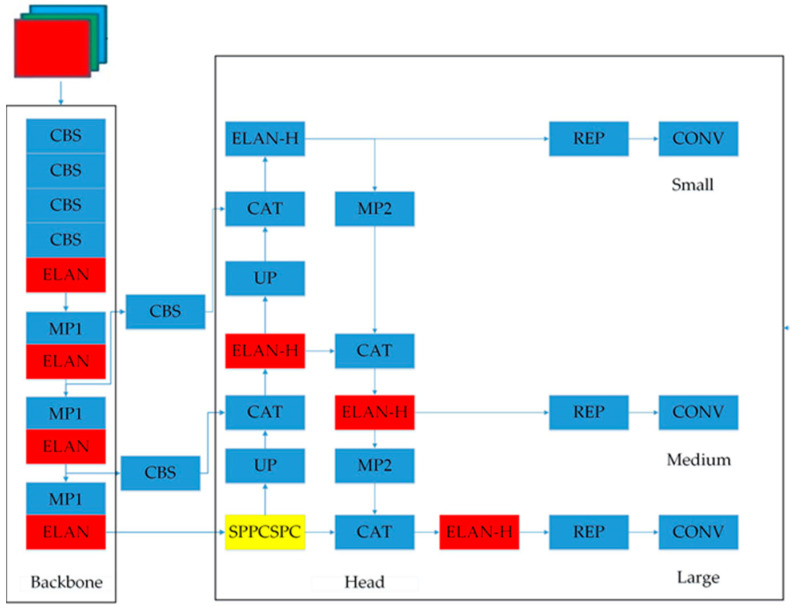
YOLOv7 model structure.

**Figure 2 sensors-23-05140-f002:**
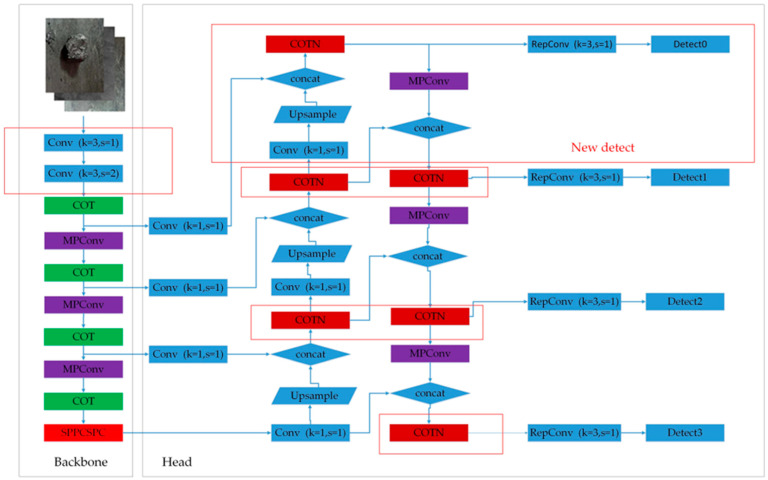
YOLOv71 + COTN model structure.

**Figure 3 sensors-23-05140-f003:**
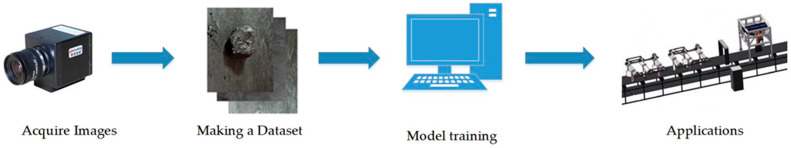
Testing methodology process.

**Figure 4 sensors-23-05140-f004:**
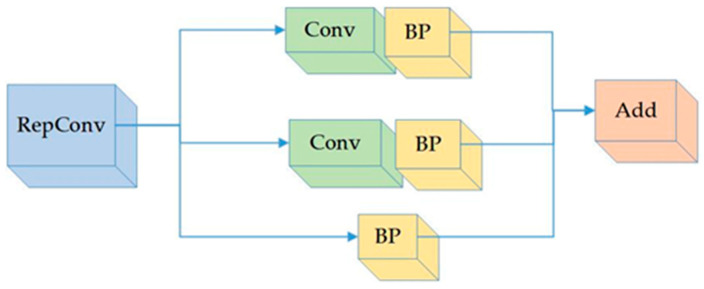
Re-parameterization components.

**Figure 5 sensors-23-05140-f005:**
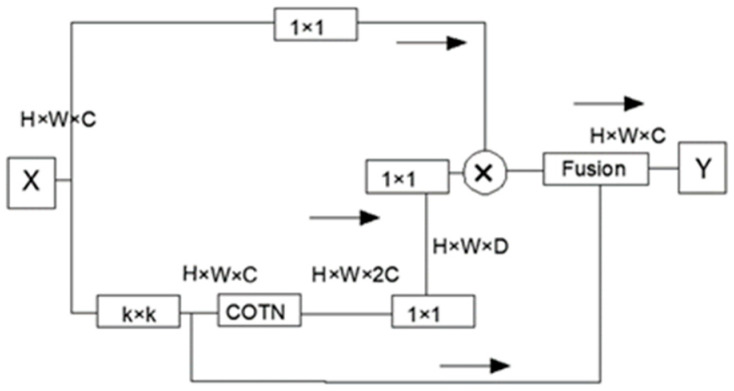
CONT module.

**Figure 6 sensors-23-05140-f006:**
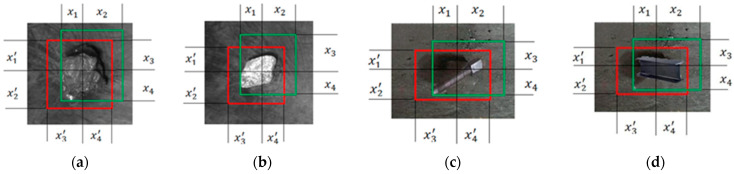
Data calculation for predicted and true boxes for each detection target. (**a**) Coal; (**b**) gangue; (**c**) bolt; (**d**) I-bean.

**Figure 7 sensors-23-05140-f007:**
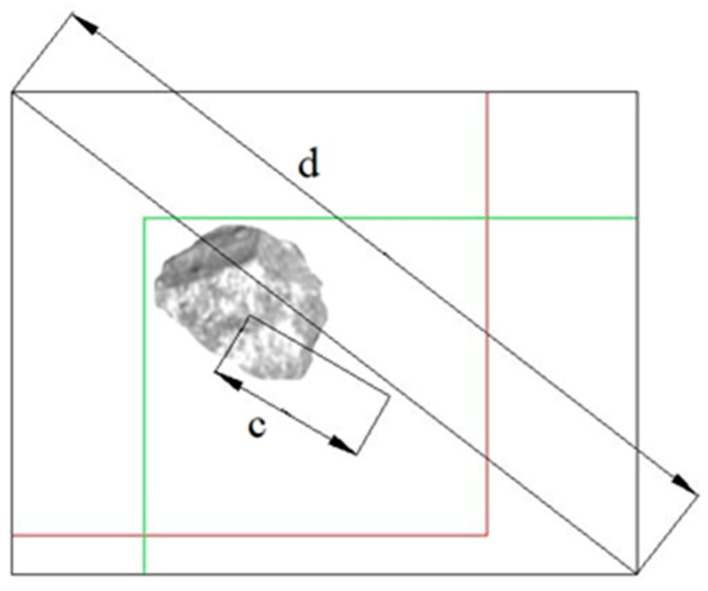
DIoU loss diagram (example of gangue).

**Figure 8 sensors-23-05140-f008:**
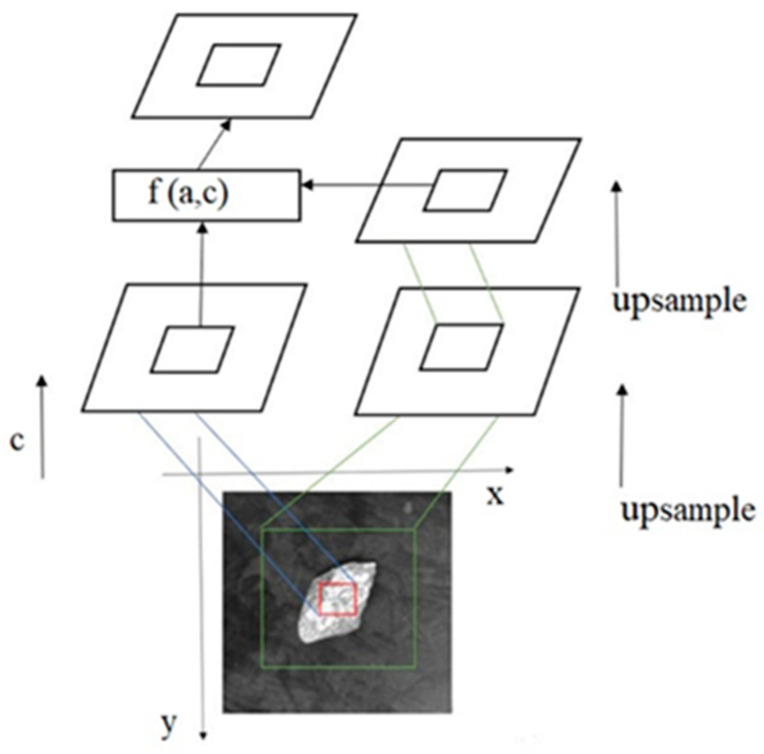
Feature extraction map (example of gangue).

**Figure 9 sensors-23-05140-f009:**
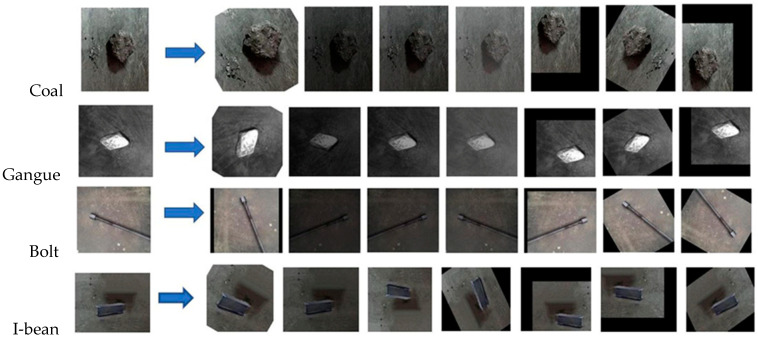
Dataset augmentation process.

**Figure 10 sensors-23-05140-f010:**
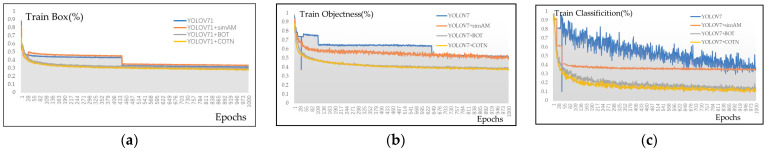
(**a**) is a plot of the results of each network model train box, (**b**) is a plot of the results of each network model train objectness, (**c**) is a plot of the results of each network model train classification.

**Figure 11 sensors-23-05140-f011:**
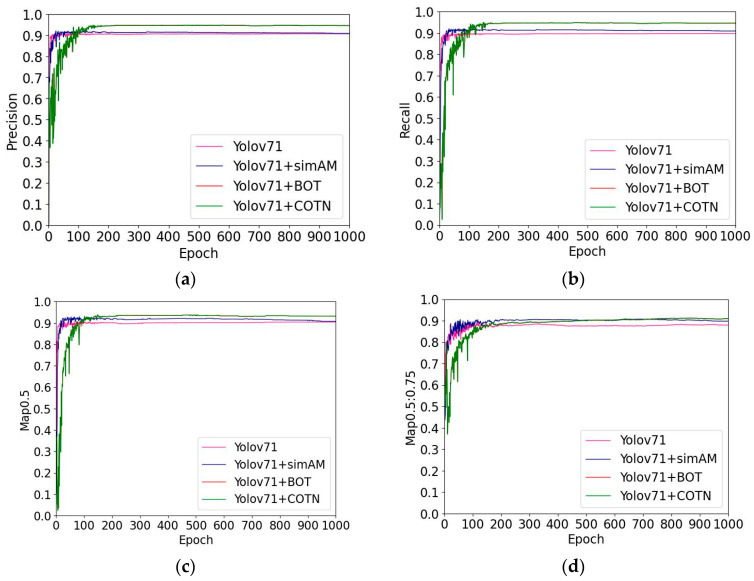
(**a**) is the accuracy of each network model run, (**b**) is the recall of each network model run, (**c**) is the mAP0.5 of each network model run, (**d**) is the mAP0.5:0.75 of each network model run.

**Figure 12 sensors-23-05140-f012:**
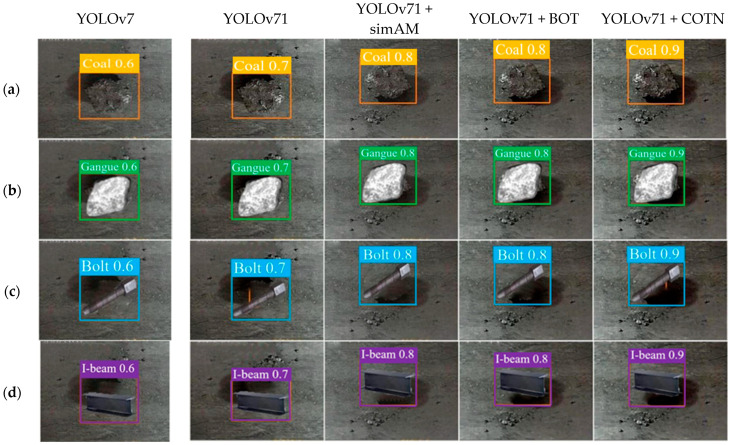
Detection of each model for the same target. From top to bottom, (**a**) is coal, (**b**) is gangue, (**c**) is bolt, (**d**) is I-beam. From left to right, they are Yolov7 network model, YOLOv71 network model, YOLOv71 + simAM network model, YOLOv71 + BOT network model, and YOLOv71 + COTN network model.

**Figure 13 sensors-23-05140-f013:**
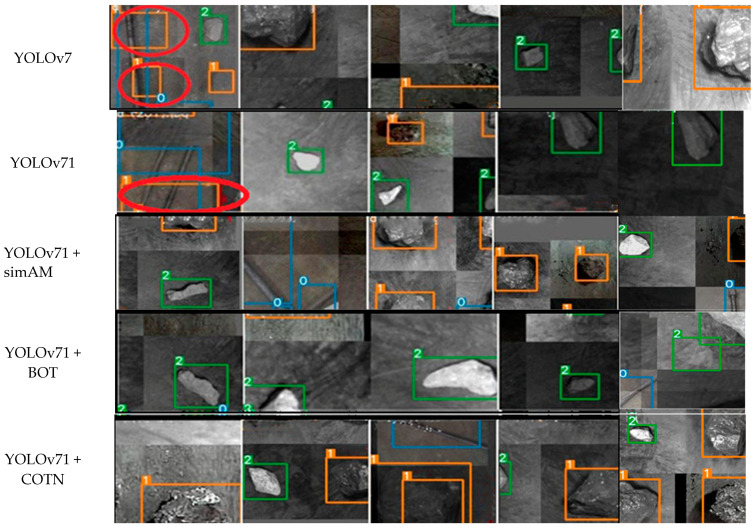
Test results for each model. From top to bottom, they are the Yolov7 model, the YOLOv71 model, the YOLOv71 + simAM model, the YOLOv71 + BotNet model, and the YOLOv71 + COTN model.

**Table 1 sensors-23-05140-t001:** Yolov7 network model parameters and Yolov71 network model parameters.

Model	Layers	Time (h)
YOLOv7	415	48.5
YOLOv71	393	45.2

**Table 2 sensors-23-05140-t002:** Model runtime environment configuration.

Designation	Version
GPU	NVIDIA Tesla T4(8G)
CUDA	10.2
Python	3.9.0
Torch	1.10.0
Torchvision	0.11.0
Torchaudio	0.10.0

**Table 3 sensors-23-05140-t003:** Model runtime parameters and module configuration.

Model	Epoch	Weight (M)	Rate of Learning	Module
YOLOv5	1000	14.4	0.01	
YOLOv7	1000	72	0.01	
YOLOv71	1000	72	0.01	
YOLOv71 + simAM	1000	72	0.01	simAM
YOLOv71 + BOT	1000	72	0.01	BOT
YOLOv71 + COTN	1000	72	0.01	COTN

**Table 4 sensors-23-05140-t004:** Operating parameters for each model.

Model	Layers	Parameters	Size	GFLOPS (G)	Gpu_mem (G)
YOLOv5	191	7,260,490	640 × 640	16.9	7.13
YOLOv7	415	37,207,344	640 × 640	105.1	3.2
YOLOv71	393	35,506,261	640 × 640	75.5	3.05
YOLOv71 + simAM	343	28,967,472	640 × 640	35.3	3.11
YOLOv71 + BOT	281	5,971,720	640 × 640	14.8	1.87
YOLOv71 + COTN	306	6,636,424	640 × 640	14.3	1.26

**Table 5 sensors-23-05140-t005:** Test results for each test target.

Model	Object	Precision	Recall	AP (0.5)	AP (0.5:0.75)
YOLOv71	Bolt	90%	87.8%	90.7%	85.7%
	Coal	88.7%	88.9%	89.7%	83.7%
	Gangue	85.8%	85.8%	85.7%	80.8%
	I-beam	84.8%	85.1%	84.5%	79.8%
YOLOv71 + simAM	Bolt	89.3%	87.5%	89.2%	84.3%
	Coal	89.6%	88.9%	89.7%	84.4%
	Gangue	87.4%	84.6%	84.6%	80.3%
	I-beam	86.4%	84.3%	84.2%	79.3%
YOLOv71 + BOT	Bolt	90%	90%	89.5%	84.9%
	Coal	89.7%	87.9%	88.6%	83.8%
	Gangue	87.5%	85.4%	85.7%	81.6%
	I-beam	86.6%	85.6%	86.9%	80.6%
YOLOv71 + COTN	Bolt	92%	92%	91.6%	86.4%
	Coal	91.8%	91.7%	91.1%	85.7%
	Gangue	90.1%	88.1%	89.3%	83.7%
	I-beam	86.9%	87.2%	87.5%	82.7%

**Table 6 sensors-23-05140-t006:** Test results for each network model.

Model	simAM	BOT	COTN	Precision	Recall	Map0.5	Map0.5:0.75
YOLOv7				87%	86.2%	87.4%	81.7%
YOLOv71				87.33%	86.9%	87.65%	82.5%
YOLOv71 + simAM	✓			88.18%	86.33%	86.93%	82.08%
YOLOv71 + BOT		✓		88.45%	87.23%	87.68%	82.73%
YOLOv71 + COTN			✓	91.3%	90.6%	91.9%	85.4%

## Data Availability

The data used to support the findings of this study are available from the writing author upon request.

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
