# Peer review of "Improved YOLOv7 Network Model for Gangue Selection Robot for Gangue and Foreign Matter Detection in Coal"

_sensors, 2023, doi:10.3390/s23115140_

Round 1

Reviewer 1 Report

The manuscript of sensors-2361677 is to propose a method of detecting gangue and foreign matter in coal by gangue selection robot with improved YOLOv7 network model. The manuscript may be interesting and can be accepted in Sensors after minor revision.

1. In Introduction,

(1) In the first four paragraphs, the corresponding literatures should be cited.

(2) The advantages and disadvantages of some traditional gangue selection methods are described in paragraph 2 in Introduction. However, where do these descriptions come from?

(3) There are many mistakes in grammar.

2. Table 3 is also “Model runtime environment configuration”?

3. The English writing may be poor. The English language must be checked and improved.

The English language must be checked and improved by Edit Service.

Author Response

Dear Reviewers:

Thank you for your valuable comments, I have revised the paper in accordance with the revisions, including the following:

1. In the introduction section

    (1) The appropriate references has been added to the first four paragraphs of the introduction.

    (2) The advantages and disadvantages of the traditional gangue screening method described in paragraph 2 of the introduction are partly from relevant references and partly from field research studies in coal mines.

    (3) Corrections were made to grammatical errors in the article.

2. Table 3 is the model runtime parameters and module configuration, the name of Table 3 has been changed to "Model runtime parameters and module configuration".

3. The whole article has been touched up in English, and the grammar and sentence structure have been improved and corrected.

4. The English has been checked and improved through specialized editing services.

5. English grammar checks and corrections to the manuscript have been marked using the “track changes” function, and changes can be displayed by clicking on the left side of the page.

6. Added a keyword "gangue selection robot".

Thank you again for your valuable comments.

Best regards,

Dengjie Yang

Reviewer 2 Report

In this paper, an improved YOLOV7 based method is proposed to detect gangue and foreign matter in coal. The layer number is reduced, and a small detection layer is added to enhance the learning ability of small object. However, multiple problems might need to be considered:

1. the motivation of proposed method should be described more clearly. 

2. How did the author define small object?

3. The experiment can not establish the effectiveness of small object detection. 

4. There are multiple writing and gramma mistakes in this paper, the author should check the whole paper more clearly.

The comments are attached in above comments.

Author Response

Dear Reviewers:

Thank you for your valuable comments, I have revised the paper in accordance with the revisions, including the following aspects:

1. The current target detection of gangue using deep learning is similar in size to that of coal blocks, with a size greater than 50mm, which is limited by the low visibility and dusty environment of the coal mine, and the detection and recognition accuracy of gangue blocks smaller than 50mm is low. In the previous detection, gangue blocks and foreign objects are not put together for simultaneous detection. This study is based on the YOLOv7 network structure for improvement, adding small target detection layer, optimising the data set, adding COTN module, improving the recognition rate of small gangue blocks and improving the recognition rate of gangue blocks and foreign objects in the coal mine environment.

2. Define gangue blocks smaller than 50mm as small objects. Paragraph 5 has been added in the first subsection of Chapter 3 to explain them.

3. This experiment was conducted in the context of the actual coal mine environment, where small gangue blocks were mixed in the coal pile above the conveyor belt. Deep learning has a high accuracy rate for large gangue blocks recognition, which is limited by the harsh environment of the coal mine, and a low recognition rate for small gangue blocks. By improving the network model, the recognition rate of gangue blocks in the coal pile is improved, that is, the recognition rate of small gangue blocks is improved.

4. The whole article has been touched up in English, and the grammar and sentence structure have been improved and corrected in the whole article.

5. English has been checked and improved through specialized editing services.

6. English grammar checks and corrections to the manuscript have been marked using the “track changes” function, and changes can be displayed by clicking on the left side of the page.

7. Added a keyword "gangue selection robot"。

Thank you again for your valuable comments.

Best regards,

Dengjie Yang
